# Inhibition of TNF-α Restores Muscle Force, Inhibits Inflammation, and Reduces Apoptosis of Traumatized Skeletal Muscles

**DOI:** 10.3390/cells11152397

**Published:** 2022-08-03

**Authors:** Ioannis Stratos, Ann-Kathrin Behrendt, Christian Anselm, Aldebarani Gonzalez, Thomas Mittlmeier, Brigitte Vollmar

**Affiliations:** 1Department of Orthopaedic Surgery, Julius-Maximilians University Wuerzburg, 97074 Wuerzburg, Germany; 2Department of Trauma, Hand and Reconstructive Surgery, University of Rostock, 18057 Rostock, Germany; behrendt.ak@googlemail.com (A.-K.B.); al.gonzalez@hotmail.it (A.G.); thomas.mittlmeier@med.uni-rostock.de (T.M.); 3Institute for Experimental Surgery, University of Rostock, 18057 Rostock, Germany; christian.alexander.anselm@gmail.com (C.A.); brigitte.vollmar@med.uni-rostock.de (B.V.)

**Keywords:** muscle injury, regeneration, infliximab, tumor necrosis factor alpha

## Abstract

Background: Muscle injuries are common in humans and are often associated with irrecoverable damage and disability. Upon muscle injury, TNF-α signaling pathways modulate the healing process and are predominantly associated with tissue degradation. In this study we assumed that TNF-α inhibition could reduce the TNF-α-associated tissue degradation after muscle injury. Materials and methods: Therefore, the left soleus muscle of 42 male Wistar rats was injured using a standardized open muscle injury model. All rats were treated immediately after injury either with infliximab (single i.p. injection; 10 mg/kg b.w.) or saline solution i.p. Final measurements were conducted at day one, four, and 14 post injury. The muscle force, the muscle cell proliferation, the muscle cell coverage as well as the myofiber diameter served as read out parameters of our experiment. Results: Systemic application of infliximab could significantly reduce the TNF-α levels in the injured muscle at day four upon trauma compared to saline treated animals. The ratio of muscle weight to body weight was increased and the twitch muscle force showed a significant rise 14 days after trauma and TNF-α inhibition. Quantification of myofiber diameter in the penumbra zone showed a significant difference between both groups at day one and four after injury, indicated by muscle hypertrophy in the infliximab group. Planimetric analysis of the injured muscle at day 14 revealed increased muscle tissue fraction in the infliximab group compared to the control animals. Muscle cell proliferation did not differ between both groups. Conclusions: These data provide evidence that the TNF-α blockade positively regulates the restauration of skeletal muscles upon injury.

## 1. Introduction

Muscle regeneration after injury is associated with a variety of pathophysiological changes. During the acute phase after trauma, processes including inflammation, angiogenesis, proliferation, apoptosis and tissue formation are occurring [1]. Inflammation extends during the first days after injury and is predominantly responsible for the degradation of necrotic tissue [1].

The Tumor Necrosis Factor alpha (TNF-α) is a crucial agent that modifies the immune response and the inflammatory-cell-induced-tissue-damage [2]. TNF-α acts directly on muscle cells and induces muscle cell protein loss and myofiber atrophy. This process of degradation and protein loss is attributed via the nuclear factor kappa-light-chain-enhancer of activated B cells (NF-κB) protein complex [3]. TNF-α can be produced and secreted by several cells of the immune system and the nervous system [4]. In the traumatized muscle, TNF-α release occurs predominantly via macrophages that infiltrate the injured tissue. Additionally, apoptotic and necrotic effects initiated by TNF-α during skeletal muscle injury negatively affect the contractile properties of the injured muscle [4]. Long-term release of TNF-α continuously activates the NF-κB signaling cascade, which forms a vicious cycle that results in a chronic inflammation [5].

Many reports associate the pro-inflammatory cytokine TNF-α with the pathogenesis of local tissue inflammation after muscle injury [6]. The TNF-α stimulates the production of acute phase cytokines plus growth factors and promotes inflammatory cell migration into the site of injury that in turn induces the repair of traumatized tissue [7]. A controversy of the TNF-α action in the skeletal muscle is summarized in the review of Zhou et al.: TNF-α pathways induce muscle degradation during muscle wasting conditions and the same TNF-α pathways are responsible for muscle regeneration in muscle injury conditions [8]. Furthermore, TNF-α is associated with various inflammatory diseases beyond the musculoskeletal system. Bronchial asthma, inflammatory bowel disease, rheumatoid arthritis, psoriatic arthritis, juvenile arthritis, and ankylosing spondylitis are common disorders that involve a TNF-α associated pathogenesis [9]. To inhibit the TNF-α induced inflammatory response, neutralization of TNF-α has been used as an effective therapy for the above diseases [10].

In this study, we assumed that TNF-α inhibition could reduce the TNF-α-associated tissue degradation after muscle injury. We hypothesized that immediate TNF-α inhibition after injury reduces excessive post-traumatic inflammatory response of the muscle and results in faster functional restoration and accelerated regeneration.

## 2. Materials and Methods

### 2.1. Experimental Groups

The current animal study was permitted by local authorities (LALLF M-V/TSD/7221.3-1.1-048/12). All experiments were performed in accordance with institutional as well as national guidelines. Therefore, we used 42 male Wistar rats (325–375 g body weight, Charles River Laboratories, Sulzfeld, Germany). During the entire experimental period, all animals were caged individually and had unrestricted access to dry chow and water.

In this setup we included two animal groups: (a) the infliximab group (IFX, *n* = 21), i.e., rats with muscle injury and consecutive i.p. application of the TNF-α inhibitor infliximab (10 mg/kg b.w. i.p.; Remicade^®^, Janssen Biotech Inc., Horsham, PA, USA) at day 0; and (b) the control group (CTRL, *n* = 21), i.e., rats with muscle injury and consecutive i.p. application of 0.9% NaCl (0.6 mL/kg b.w. NaCl i.p.) at day 0.

Final investigations were conducted at experimental day one, day four or day 14 after muscle injury (*n* = seven animals per time point and group). After muscle strength measurement, the injured left soleus muscle was excised and appropriately stored for further analysis (weight measurement, histology, and immunohistochemistry).

To analyze the muscle cell proliferation, Bromdesoxyuridin (BrdU) was used. Therefore, all animals were treated 48 h prior to tissue harvest with a single i.p. injection of BrdU (50 mg/kg). Animals were euthanized via increasing the anesthesia after tissue harvest was completed.

### 2.2. Induction of Muscle Injury

All animals (*n* = seven animals per time-point and group) were anesthetized, and the left soleus muscle was exposed and injured as described previously [11]. Briefly, the muscle injury was conducted using a clamp equipped with strain gauges. The soleus was injured by applying the clamp over its total muscle surface (70 mm^2^; applied force 25 N for 10 s) excluding the tendons of the muscle and the central area with the muscle-supplying nerves and vessels. After injury induction, the exposed region was closed layer-by-layer.

### 2.3. Muscle Strength Measurement

Depending on the experimental group, the muscle strength measurement was performed at day one, day four or day 14 after muscle injury. Therefore, animals (*n* = seven animals per time-point and group) underwent anesthesia, and both sciatic nerves, as well as the left (injured) and right (non-injured) soleus muscle, were exposed. The Achilles’ tendon was separated from the calcaneal bone and connected to a digital force gauge. By stimulating the sciatic nerve with 9 mA/75 Hz we were able to derive a fast twitch force (duration of neve-stimulation: 0.1 s) or a tetanic contraction (duration of neve-stimulation: 3 s) of the soleus muscle. For the calculation of fast twitch and tetanic contraction we used the first five curves and computed the mean of the maximal values of each curve as described in our previous work [11].

### 2.4. Immunohistochemistry

For immunohistochemical analysis of the injured soleus muscle (*n* = seven animals per time-point and group), 10 µm paraffin slides were prepared and incubated with antibodies. As chromogen we used 3,3′-diaminobenzidine. For the detection of BrdU, paraffin slides were incubated with a mouse anti-BrdU mAb (1:50; No. M0744, Dako Cytomation, Hamburg, Germany). For the detection of NF-κB p65, paraffin slides were incubated with a rabbit anti-NF-κB mAb (1:100; No. SC-372G, Santa Cruz Biotechnology, Santa Cruz, CA, USA).

### 2.5. Histology

For the histology of the injured soleus muscle (*n* = seven animals per time-point and group), 10 µm paraffin slides were used.

Apoptotic cells were detected using the ApopTag kit (ApopTag, S7100, Chemicon, Temecula, CA, USA) by the TUNEL method. The kit was applied as recommended in the instruction’s manual.

The chloroacetate esterase (CAE) staining was used to identify tissue-infiltrating leukocytes (mainly neutrophiles and mast cells). The slides were therefore deparaffinized and treated with naphtol AS-D chloroacetate (Sigma Diagnostics, Deisenhofen, Germany). The slides were then dehydrated in an ascending alcohol series. Infiltrating leukocytes were visualized by imaging the naphtol compound through hydrolysis of the naphtol AS-D chloroacetate by leukocyte-specific esterases.

The hematoxylin-eosin (HE) stained slides were used to visualize the integrity of the injured muscle tissue. For the HE-staining, the paraffin slides were deparaffinized, stained with Mayers hematoxylin solution, rinsed, counterstained with Eosin solution, dehydrated and cleared using standard laboratory techniques.

### 2.6. Quantification of the Histology and Immunohistochemistry

All stained sections were analyzed using a light microscope (BX 51, Olympus, Hamburg, Germany), with a ×40 objective and a numerical aperture of 0.65. BrdU, TUNEL and CAE positive cells were counted between the two insertions of the muscle (approximately 30 sequential fields of view). For the NF-κB staining, only NF-κB positive nuclei localized within muscle cells or in direct vicinity to the muscle cell were counted. Thereby, NF-κB positive muscle cell nuclei or NF-κB positive satellite cells were counted. The quantification was carried out in 10 observation fields of the penumbra zone (i.e., the muscle tissue next to the injury). Final values are given as positive cells per mm^2^.

For the quantitative analysis of the HE-staining, we used high-resolution digital images of the muscle (Olympus Colorview II) and planimetrically measured the visible muscle tissue (Photoshop CS2, Adobe, Dublin, Ireland). Using the HE-staining we also quantified the myofiber diameter in the penumbra zone (0.01 mm^2^; 4 observation fields; ×20 objective) using the light microscope software Cell^D (Olympus, Hamburg, Germany). All methods are previously described by our group [11].

### 2.7. Western Blot Analysis

The protein expression of TNF-α, TNF receptor associated factor 2 (TRAF-2), nuclear factor kappa-light-chain-enhancer of activated B cells subunits p65 and p105 (NF-ĸB p65 and p105), interleukin 6 (IL-6) and glyceraldehyde 3-phosphate dehydrogenase (GAPDH) as loading control were quantified by means of western blot.

Therefore, the injured muscle (*n* = four animals per time-point and group) was prepared for the Western blot analysis (homogenization, lysis, incubation on ice and centrifugation for 10 min at 10,000× *g*). The BCA assay was used to quantify proteins using bicinchoninic acid with bovine serum albumin (BSA) as standard (Pierce Biotechnology, Rockford, IL, USA). Per lane 20 µg (TNF-α, TRAF-2, NF-ĸB,) or 60 µg (IL-6) protein extract was separated discontinuously on 6%, 10% (TRAF-2, NF-ĸB p65 or p105) and 14% (TNF-α, IL-6) SDS polyacrylamide gels and transferred to a polyvinylidifluoride membrane (Immobilon-P transfer membrane; Millipore, Billerica, MA, USA). Nonspecific binding sites were blocked with 2.5% BSA (Biotechnology, Santa Cruz, CA, USA), and membranes were incubated over night at 4 °C with primary antibodies followed by secondary antibody incubation for 1 h at RT.

The following antibodies were used: TNF-α (#sc-1350, 1:1000; secondary Ab: donkey anti-goat IgG HRP, #sc-2020, 1:10,000, Santa Cruz Biotechnology, Santa Cruz, CA, USA), TRAF-2 (#ab126758, 1:2.000, Abcam, Cambridge, UK; secondary Ab: goat anti-rabbit IgG HRP, #sc-2004, 1:20,000, Biotechnology, Santa Cruz, CA, USA), NF-ĸB (p65 #sc-372G, 1:500 Biotechnology, Santa Cruz, CA, USA; p105 #14-6732, 1:500, eBioscience; secondary Ab: donkey anti-goat IgG HRP, #sc-2020, 1:40,000, Biotechnology, Santa Cruz, CA, USA; anti-rabbit IgG HRP, #7074, 1:20,000 Cell Signaling, Danvers, MA, USA), IL-6 (#sc-1265, 1:1000; secondary Ab: goat anti-rabbit IgG HRP, #sc-2004, 1:10,000, Biotechnology, Santa Cruz, CA, USA).

Protein expression was visualized by chemiluminescence reaction (ECL Plus; Amersham Pharmacia Biotech, Freiburg, Germany), digitized using ChemiDocTM XRS System (Bio-Rad Laboratories, Munich, Germany) and normalized to the GAPDH signals (#MAB374, 1:20,000, Millipore, Billerica, MA, USA; secondary Ab: anti-mouse IgG-Peroxidase, #A9044, 1:40,000, Sigma-Aldrich, Hamburg, Germany).

### 2.8. Statistical Analysis

All values were given as mean ± standard error of the mean (SEM). Prior to statistical analysis, all data were tested for normality. If normal distribution was present, differences between groups were examined using an unpaired Student’s t-test, otherwise a Mann-Whitney U test was applied. The software SigmaStat 3.0 (Jandel Corporation, San Rafael, CA, USA) was used for statistical analysis. Statistical significance was assumed at *p* < 0.05 for all tests. We used SEM to provide a measure of precision of the estimated mean of the analyzed population.

## 3. Results

All animals woke up from the anesthesia without any complications. During the experimental period, no signs of illness or discomfort were noted. All animals gained normal weight during the experimental period (on average 3.1 g b.w./d; no difference between groups).

The non-injured right soleus muscle showed a normal contractility as shown by the twitch force in N (CTRL group: 0.42 ± 0.06 [d1], 0.44 ± 0.08 [d4], 0.52 ± 0.02 [d14]; IFX group: 0.51 ± 0.09 [d1], 0.46 ± 0.09 [d4], 0.54 ± 0.03 [d14]) and the tetanic contraction in N (CTRL group: 1.05 ± 0.07 [d1], 0.98 ± 0.15 [d4], 1.23 ± 0.06 [d14]; IFX group: 1.24 ± 0.23 [d1], 1.10 ± 0.13 [d4], 1.29 ± 0.09 [d14]). No statistical difference in the muscle force of the right (non-injured) soleus muscle was observed between the timepoints and groups. A substantial loss of twitch force and tetanic contraction was detected immediately after muscle injury on day 1 in the CTRL group (twitch force: 0.04 ± 0.01, tetanic force: 0.09 ± 0.04 N) and the IFX group (twitch force: 0.10 ± 0.07, tetanic force: 0.16 ± 0.04 N). From day four up to day 14, the muscle force showed a continuous improvement in the control group. The recovery of the injured soleus in the CTRL group at later time points could not fully restore the muscle strength. In the CTRL group, at day 14 after injury, the twitch force was 0.29 ± 0.02 N and the tetanic force was 0.67 ± 0.09 N. The application of infliximab after injury resulted in enhanced muscle force, as indicated by the significant increase in twitch muscle force on day 14 (0.39 ± 0.04 * N) and higher mean values for the tetanic force (0.80 ± 0.08 N) compared to CTRL (Figure 1A,B).

A peak of infiltrating leukocytes in the injured tissue was observed in the control group at day one after injury, comprising 156 cells/mm^2^ (Figure 2A). The local inflammatory process diminished with time until day 14 involving only 11 cells/mm^2^. Animals from the infliximab group showed a significant reduction of CAE positive cells including an average 122, 49 and 7 cells/mm^2^ for days one, four, and 14, respectively (*p* < 0.001 vs. control group) (Figure 2A). A similar kinetic was observed for the apoptotic cell death, determined by the TUNEL-staining. Apoptosis reached its maximum in the control group at day four with 42 cells/mm^2^ and decreased over time to 6 cells/mm^2^ at day 14 after injury (Figure 2B). Treatment with the TNF-α inhibitor infliximab significantly reduced the apoptotic cell death at each time point when compared to the CTRL group (Figure 2B).

The planimetric analysis of the muscle showed reduced muscle debris upon injury and infliximab treatment on day 14 compared to CTRL (Figure 3A). Quantification of the myofiber diameter in the penumbra zone showed a significant reduction in the control group on days one and four compared to the infliximab group (Figure 3B). Since healthy uninjured myofibers have a diameter between 38 and 42 µm [12], our results indicate that the post-traumatic muscle atrophy during days one and four can be prevented by infliximab treatment. Additionally, treatment with infliximab caused a significant increase in muscle mass 14 days post muscle injury, as implied by the enhanced ratio of muscle weight to body weight in the infliximab group compared to the control group (Table 1). The non-injured muscle did not show any difference in its muscle mass with or without infliximab treatment (Table 1). Analysis of cell proliferation using BrdU immunohistochemistry did not show any difference between the groups at any time point (Figure 4A).

To elucidate the activity of inflammatory cytokines in the injured muscle tissue, we analyzed the levels of TNF-α and IL-6. On day 4, the TNF-α blocker infliximab could significantly reduce the TNF-α levels in the injured muscle compared with the TNF-α levels of the CTRL group (Table 2 and Figure 5). The densitometric analysis for IL-6 did not show any difference with or without infliximab treatment (Table 2 and Figure 5).

Furthermore, we measured the levels of TRAF-2, a signaling adaptor molecule crucially involved in the initiation of TNF-α receptor signaling cascade leading to NF-κB up-regulation. Muscle injury resulted in an up-regulation of TNF-α signaling since TRAF-2 levels were found increased four and 14 days after injury in the IFX as well as in the CTRL group (Table 3). Infliximab did not regulate the TRAF-2 protein levels, as given by comparable densitometric values of the ratio TRAF-2 to GAPDH in both groups at all time points (Table 3). Immunohistochemical analysis of NF-κB p65 levels in the penumbra area showed significant changes between both groups at all time points (Figure 4B). Infliximab caused a significant decrease of NF-κB p65 positive nuclei at days 1 and 14, whereas at day four a significant increase was observed compared to the control group. On the contrary, quantification by Western blot of p65 and p105 as NF-κB subunits in the entire injured muscle tissue showed no changes in both groups and all time points (Table 3 and Figure 5).

## 4. Discussion

We showed that the functional restoration of the muscle after injury and IFX treatment is associated with increased muscle force and muscle tissue at day 14, as well as a decrease of apoptosis and local leukocyte infiltration throughout the first 14 days. Additionally, IFX treatment increases muscle tissue area and muscle volume compared with untreated injured controls.

### 4.1. Methodological Considerations: Dose and Pharmacokinetics of Infliximab

In this study, we treated animals immediately after injury with infliximab administered intraperitoneally at a concentration of 10 mg/kg body weight. Once infliximab is systemically applied, it distributes in the intravascular space with an elimination half-life time between seven and 12 days and a mean residence period of up to 17 days [13]. Concerning the concentration of infliximab, numerous studies have reported beneficial effects on humans, rats, and mice in similar quantity that we used. The usual therapeutic dose in humans for Crohn’s disease or rheumatoid arthritis is 3–10 mg/kg and is applied by means of intravenous infusions [13,14]. A similar concentration is used in most animal studies. Protective effects have been demonstrated in diabetes [15], muscle graft inflammation [16], and osteoporosis [17] models by a single or weekly application of infliximab at a concentration between 10 and 20 mg/kg body weight. Therefore, we believe that the used IFX dose is sufficient to induce biological effects in rats.

### 4.2. The Control Group

Although the use of saline as a control solution is common for infliximab studies, a more appropriate control solution would be an isotype matched control antibody. In the case of infliximab, the appropriate monoclonal control antibody would be an immunoglobulin G1 (IgG1). Regatieri et al. compared the effects of intravenous balanced salt solution, monoclonal IgG1 antibodies and infliximab application in an experimental choroidal neovascularization rat model, showing an almost similar choroidal neovascularization between the balanced salt solution and the IgG1 group [18]. The influence of isolated immunoglobulins on the skeletal muscle remains controversial: reports show [19] positive effects of immunoglobulins in muscle diseases, and other studies do not show [20] any benefit after immunoglobulin treatment in the same diseases. For that reason, we assume that the bias resulting from our saline control group compared with an IgG1 control group is neglectable.

### 4.3. Actions of TNF-α and Infliximab

Upon muscle injury, local leukocytes infiltrate the damaged tissue. The Leukocytes’ task is to regulate the recovery of traumatized muscle and to participate in the regeneration by balancing tissue breakdown and muscle restoration. The inflammatory process begins immediately after injury and ends 10–14 days later. During the inflammatory process, neutrophiles control the initial post-traumatic phase by phagocytosing cells. Macrophages and later lymphocytes mediate the secondary and tertiary tissue inflammation after injury [21]. Similarly, apoptotic events occur during the first week after injury and regulate the number of residual cells in the damaged tissue. Infliximab reduces the excessive local inflammatory response during the first 14 days after injury and increases the cell-quantity after injury, predominantly on day 4. These events contribute to an increased cellular turnover that in turn results in functional restoration (indicated by the increased muscle strength) of the damaged tissue.

Various in vitro experiments with muscle cells have shown that TNF-α exhibits diverse effects on myogenesis. It has been reported that, depending on the state of muscle cell differentiation, TNF-α promotes muscle cell death, muscle cell growth, or even muscle wasting [22]. In vivo studies illustrate difficulties in predicting the effects of TNF-α on muscle cell growth and viability because positive and negative regulatory effects of TNF-α on the muscle tissue have been reported. The degenerative effects of TNF-α on muscle tissue are mediated by the stimulation of muscle atrophy as well as via decreased protein production and increased protein degradation, respectively. This occurs via the proteolytic ubiquitin-proteasome pathway, which in turn leads to the upregulation of NF-κB [23]. In contrast to previous findings, Li et al. observed that increased systemic levels of TNF-α activate satellite cells that in turn enter mitosis in the skeletal muscle [24]. Concerning the previously mentioned controversy, Chen and coworkers [25] proposed that the effects of TNF-α on the injured muscle are related to TNF-α concentration. A short-term upregulation of TNF-α intensifies myogenesis, whereas the continuous release of TNF-α negatively impairs muscle regeneration [25].

Because of the complex effects of TNF-α on traumatized muscle tissue and the multiple interactions between inflammation and TNF-α blockade, it is very difficult to accurately predict the effect of infliximab after muscle injury. According to current literature, NF-κB mediated pro-inflammatory cytokines, like TNF-α and IL-6, are not activated in healthy (un-injured) muscle tissue. Additionally, no up- or down-regulation is expected in normal muscle tissue for TRAF-2, p105 and p65 proteins [26]. Our results show that infliximab influences apoptosis, inflammation, NF-κB levels and muscle tissue. Regarding the biological actions of infliximab, it is known that it attaches to the soluble and membrane-bound forms of TNF-α. Once the activity of TNF-α is blocked, numerous cellular events take place. These events result in a local and systemic pro-inflammatory cytokine downregulation, a reduction of local leucocyte infiltration, a decreased local lymphocyte count, the induction of apoptosis of TNF-α -producing cells, the reduction of acute phase proteins, and the increase of NF-κB inhibitor levels [13]. TNF-α receptor associated factors (TRAF) interact with distinct members of the tumor necrosis factor receptor (TNFR) superfamily and other transmembrane receptors [27]. This results in the assembly of larger signaling complexes initiating signaling cascades including inflammatory responses and apoptosis. Regarding the apoptotic response of tissues to infliximab, there are many papers that describe the pro-apoptotic effects of infliximab [28]. Radley et al. demonstrated that blockade of TNF-α reduces myofiber damage and necrosis of muscle cells in a dystrophic mouse model [29]. Other groups observed similar results [30] after eccentric exercise and infliximab administration in dystrophic and normal mice. Additionally, Kurt et al. reported anti-apoptotic effects on neurons after infliximab treatment in a rabbit hydrocephalus model [31]. Unfortunately, the molecular changes responsible for anti-apoptotic effects after infliximab administration are not well described and could well be the subject of future investigations.

Inflammatory pathways are essential for the regeneration of traumatized muscle tissue. Macrophages act as immunological and non-immunological regulators during muscle regeneration. They are responsible for limiting the local inflammatory response and at the same time provide a trophic function to muscle cells and local stem cells. Additionally, macrophages promote angiogenesis and enhance local remodeling of the damaged tissue [32]. In this study, we assumed an increased phagocytosis of cell fragments during the first days after injury. This statement seems likely if we consider the muscle tissue fraction parameter as the reciprocal value of the granulation tissue. Thus, treatment with infliximab on day 14 after trauma is associated with increased muscle tissue fraction, i.e., reduced granulation tissue. This increase in muscle tissue fraction may result from increased phagocytic activity at baseline.

The pleiotropic nature of the NF-κB transcription factor becomes evident considering current literature highlighting its role during skeletal muscle physiology and disease. NF-κB activity seems to directly regulate muscle differentiation, mitotic events and probably other molecules that control muscle energy and metabolism upon atrophic conditions [33]. The inhibition of NF-κB signals is a known action of infliximab [34]. It is considered that infliximab treatment increases NF-κB inhibitor levels including nuclear factor of kappa light polypeptide gene enhancer in B-cells inhibitor alpha and gamma (IκBα and IκBγ). This in turn downregulates NF-κB signals and is associated with a decreased TNF-α production [34]. In our results we could observe that infliximab did not affect muscle cell proliferation upon injury. This observation is quite reasonable based on the existing literature. It is known that IκBβ-mediated canonical NF-κB signaling regulates myogenesis in traumatized skeletal muscle, and that the targeted ablation of IκBβ reduces the number of satellite cells in injured skeletal muscle, possibly by inhibiting their proliferation and survival [35]. We assume that IFX inhibits (via the canonical NF-κB pathway) the IκBβ that in turn reduces the number of proliferating satellite cells in the injured skeletal muscle tissue. Similar effects could be observed in our study indicated by the decreased NF-κB production during day one and 14 in the myofibers and decreased TNF-α levels in the injured muscle tissue on day four. The increased levels of myocytes NF-κB at day four could be seen because of anabolic events that occur in the injured myofibers during cellular reconstruction in the phase of regeneration, as described by others [2].

There are very few reports regarding the muscle strength after infliximab treatment. Kakoulidou and coworkers [36] reported in a case report that repeated treatment with infliximab results in improved muscle fatigability tests in a patient with myasthenia gravis. Another clinical trial showed that chronic treatment with infliximab enhances heart function in patients with rheumatoid arthritis [37], indicating an improved function of the cardiomyocytes. Subramaniam et al. showed that infliximab increases muscle tissue in patients with Crohn’s disease [38]. The pathophysiological mechanism responsible for this result on the muscle is not known in detail. Piers et al. could demonstrate that TNF-α blockade could reduce the muscle force deficit in a dystrophic mouse model [30]. The same group has suggested that the protective effects of TNF-α blockade on contractile dysfunction is probably a short-term result of TNF-α-inhibition [30]. We hypothesize that the significant increase in myofiber diameter at day one and day four after injury in the penumbra zone is mainly mediated by infliximab. Infliximab protects the penumbra zone after injury and promotes muscle fiber survival during the first posttraumatic days. At later time points (day 14), no protective effect of infliximab on muscle fiber diameter can be mediated. This initial protective effect leads secondarily to an increase in the visible muscle tissue area, as observed at day 14 post-injury in the IFX group.

Recent studies have identified the crucial role of fibro/adipogenic progenitors (FAPs) during muscle regeneration and repair [39]. FAPs are an essential source of matrix-producing myofibroblasts in the skeletal muscle and are in pathological conditions responsible for muscle fibrosis and fatty degeneration. Some studies describe a negative effect of muscle regeneration after TNF-a inhibition. This argumentation is predominantly based on the action of FAPs and by extension on the fatty degeneration of the muscle. During repair, FAPs are involved in the muscle regeneration by creating a temporary niche that aids the myoblasts’ function. Almost immediately after muscle injury, infiltrating macrophages induce apoptosis in FAPs. This apoptotic process is mediated via TNF-signaling [40]. The TNF blockade leads to the increased survival of FAP and increased collagen deposition in the injured muscle tissue [40]. Marinkovic et al. provided evidence that NOTCH- and TNF-α-signaling control FAP differentiation in the skeletal muscle [41]. TNF-α, restrains FAP adipogenesis via a synergistic interaction between NF-κB and the NOTCH-signaling by favoring regeneration in the mdx muscles. In the same study, the use of anti-TNF-α antibodies enhanced the antiadipogenic effects [41].

Previous research has shown that muscle injury changes the ratio of type I and II myofibers and that the soleus muscle (“type II muscle”) transiently transforms after injury to a fast twitching muscle (“type I muscle”) [11]. In vivo experiments in mice have shown that the blockade of TNF-α prevents the degradation of slow contracting (type II) myofibers [42]. Furthermore, type II myofibers are predominantly responsible for the twitch force in the skeletal muscle [43]. We assume that the reduced degradation of type II myofibers induced by infliximab may lead to an increased twitch muscle force, as indicated by the significant increase of the twitch force in our results. Furthermore, infliximab most likely does not have a significant protective influence on type I myofibers that are responsible for the tetanic contraction.

A limitation of our study is that our results cannot be generalized for every peripheral muscle. Type-I fibers have a higher rate of protein turnover and are more vulnerable to atrophy after inactivity or denervation. Type-II fibers, on the other hand, are negatively affected by protein degradation [44]. The soleus muscle predominantly contains type-I (slow oxidative) fibers. Studies in healthy human subjects have shown that TNF-α is mainly expressed by type-II (fast oxidative and fast glycolytic) fibers [45]. Pathological conditions, including muscle wasting, appears to be TNF-α mediated and myofiber specific. Sciorati et al. showed that the proportion of type-II fibers decreases, and that type-I fibers increase during aging-induced muscle wasting. The authors discovered that the inhibition of TNF-α could reduce the loss of type-II fibers in aged animals [42]. We therefore assume that the muscle regeneration after TNF-α blockade could be more pronounced in predominantly type-II muscles than in predominantly type-I muscles.

Infliximab treatment for muscle injuries has highlighted the multiple pathways that are regulated by TNF-α inhibition. Our study unraveled some of the complex cellular and cytokine-dependent pathways that are involved in the regeneration of skeletal muscle, highlighting the importance of TNF-α and its inhibition during muscle injury and recovery.

## Figures and Tables

**Figure 1 cells-11-02397-f001:**
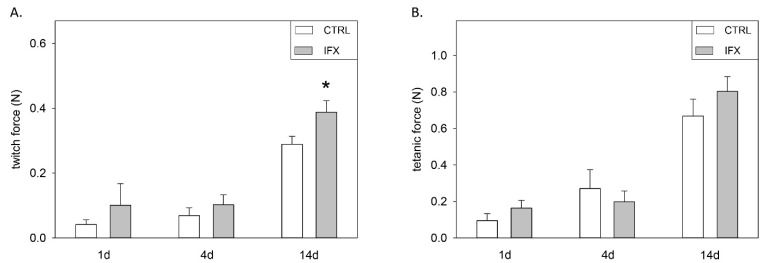
Twitch force (**A**) and tetanic force (**B**) of the injured soleus muscle. Animals received either infliximab (IFX) or NaCl (CTRL) directly after the soft tissue injury. Measurement was conducted at day one, four and 14 (means ± SEM; *n* = seven Wistar-rats per group and analyzed day; * *p* < 0.05 vs. CTRL).

**Figure 2 cells-11-02397-f002:**
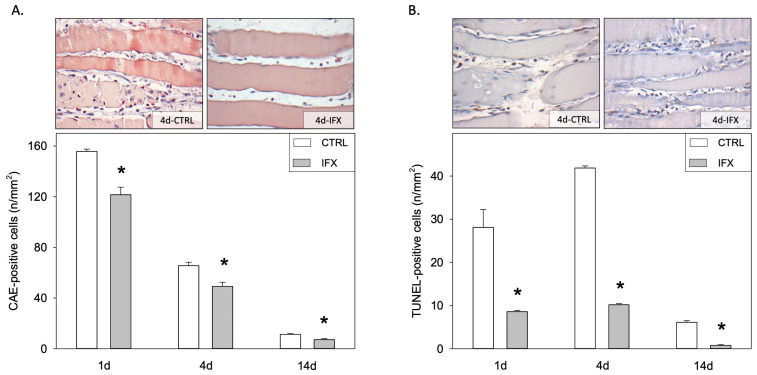
Representative light microscopic images (upper panels; day four; original magnification: ×400) and quantitative analysis (lower panels) of CAE positive cells (**A**) and TUNEL positive cells (**B**) of the injured soleus muscle. Animals received either infliximab (IFX) or NaCl (CTRL) directly after the soft tissue injury. Measurement was conducted at day one, four and 14 (means ± SEM; *n* = seven Wistar-rats per group and analyzed day; * *p* < 0.05 vs. CTRL).

**Figure 3 cells-11-02397-f003:**
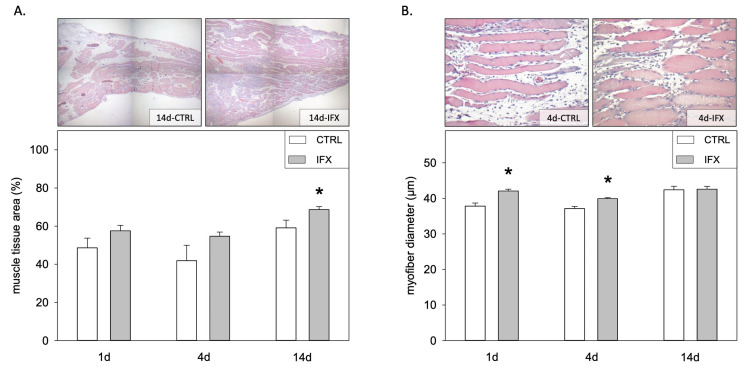
Representative light microscopic images (upper panels) and quantitative analysis (lower panels) of the injured soleus muscle (**A**) and myofiber diameter (**B**) of the penumbra zone in HE stained sections. Original magnification: ×40 at day 14 (**A**); ×200 at day 4 (**B**). Animals received either infliximab (IFX) or NaCl (CTRL) directly after the soft tissue injury. Measurement was conducted at days one, four, and 14 (means ± SEM; *n* = seven Wistar-rats per group and analyzed day; * *p* < 0.05 vs. CTRL).

**Figure 4 cells-11-02397-f004:**
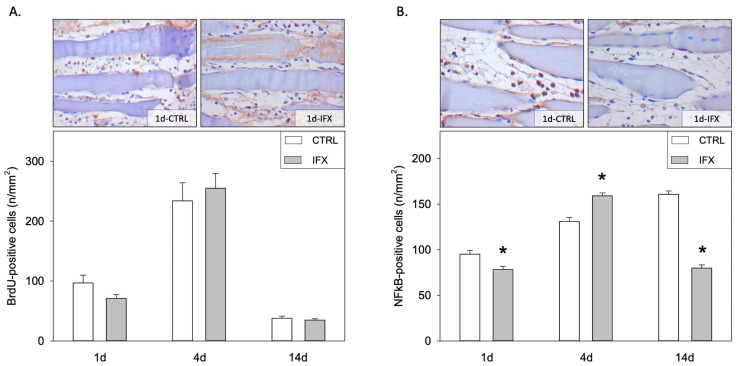
Representative light microscopic images (upper panels; day one; original magnification: × 400) and quantitative analysis (lower panels) of BrdU positive cells of the injured soleus muscle (**A**) and of NF-κB p65 positive nuclei in the penumbra area (**B**). Animals received either infliximab (IFX) or NaCl (CTRL) directly after the soft tissue injury. Measurement was conducted at days one, four, and 14 (means ± SEM; *n* = seven Wistar-rats per group and analyzed day; * *p* < 0.05 vs. CTRL).

**Figure 5 cells-11-02397-f005:**
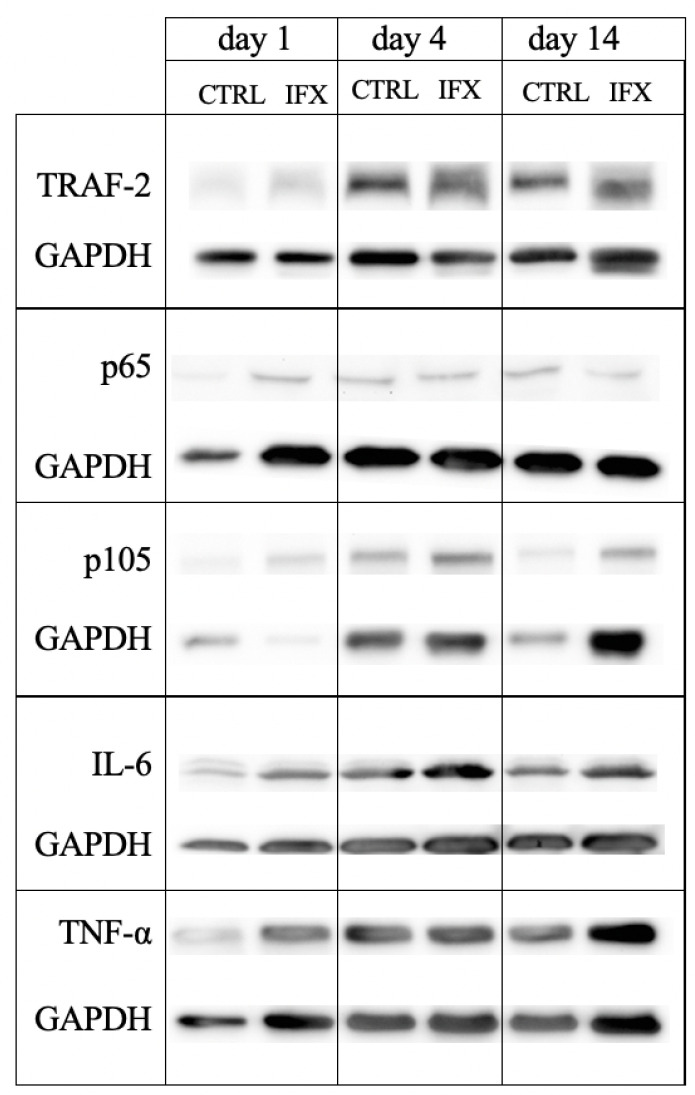
Representative Western blot of TRAF-2, NF-κB subunits p105 and p65, IL-6 and TNF-α per GAPDH protein in the injured soleus muscle tissue. Animals received either infliximab (IFX) or NaCl (CTRL) directly after the soft tissue injury. Measurement was conducted at days one, four, and 14.

**Table 1 cells-11-02397-t001:** Ratio of the muscle weight to body weight of the non-injured and injured soleus muscle.

	Ratio of the Non-Injured (Right) Soleus Muscle Weight to Body Weight (mg/g)		Ratio of the Injured (Left) Soleus Muscle Weight to Body Weight (mg/g)	
Day	CTRL	IFX	CTRL	IFX
1	0.90 ± 0.02	0.91 ± 0.10	0.89 ± 0.04	0.88 ± 0.10
4	0.78 ± 0.02	0.78 ± 0.01	0.78 ± 0.02	0.77 ± 0.01
14	0.76 ± 0.03	0.83 ± 0.02	0.71 ± 0.01	0.80 ± 0.01 *

Animals received either infliximab (IFX;) or NaCl (CTRL) directly after the soft tissue injury. Measurement was conducted at days one, four and 14 (means ± SEM; *n* = seven Wistar-rats per group and analyzed day; * *p* < 0.05 vs. CTRL).

**Table 2 cells-11-02397-t002:** Quantitative densitometric Western blot analysis of IL-6 and of TNF-α per GAPDH protein levels in the injured soleus muscle tissue.

	Ratio of IL-6 to GAPDH		Ratio of TNF-α to GAPDH	
Day	CTRL	IFX	CTRL	IFX
1	0.53 ± 0.25	0.46 ± 0.11	0.27 ± 0.11	0.38 ± 0.27
4	0.62 ± 0.16	0.46 ± 0.09	0.50 ± 0.12	0.12 ± 0.05 *
14	0.35 ± 0.05	0.46 ± 0.10	0.16 ± 0.05	0.25 ± 0.15

Animals received either infliximab (IFX) or NaCl (CTRL) immediately after injury. Measurement was conducted one, four, and 14 days after injury (means ± SEM; *n* = four Wistar-rats per group and analyzed day; * *p* < 0.05 vs. CTRL).

**Table 3 cells-11-02397-t003:** Quantitative densitometric Western blot analysis of TRAF-2, NF-κB subunits p105 and p65 per GAPDH protein levels in soleus muscle tissue.

	Ratio of TRAF-2 to GAPDH		Ratio of p105 to GAPDH		Ratio of p65 to GAPDH	
Day	CTRL	IFX	CTRL	IFX	CTRL	IFX
1	0.24 ± 0.08	0.41 ± 0.15	0.13 ± 0.05	0.07 ± 0.01	0.13 ± 0.06	0.19 ± 0.12
4	1.02 ± 0.07	0.91 ± 0.38	0.13 ± 0.01	0.12 ± 0.04	0.05 ± 0.01	0.08 ± 0.02
14	1.33 ± 0.46	0.72 ± 0.19	0.10 ± 0.02	0.10 ± 0.02	0.08 ± 0.02	0.06 ± 0.01

Quantitative densitometric Western blot analysis of TRAF-2, NF-κB subunits p105 and p65 per GAPDH protein levels in the injured soleus muscle tissue. Animals received either infliximab (IFX) or NaCl (CTRL) directly after the soft tissue injury. Measurement was conducted at days one, four and 14 (means ± SEM; *n* = four Wistar-rats per group and analyzed day; * *p* < 0.05 vs. CTRL).

## Data Availability

The data generated during current study are available from the corresponding author on reasonable request.

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
