# Peer review of "Inhibition of TNF-α Restores Muscle Force, Inhibits Inflammation, and Reduces Apoptosis of Traumatized Skeletal Muscles"

_cells, 2022, doi:10.3390/cells11152397_

Round 1

Reviewer 1 Report

The revised manuscript has responded adequately to my comments

Author Response

Dear Reviewer 1 

Thank you for accepting our primary revision. 

Reviewer 2 Report

I think that Authors satisfactorily answered to all the raised criticisms.

I suggest to carefully check possible grammar/typing errors throughout the text since I found some little mistakes such as:

line 130: was used to identify (instead of identified)

line 539: mice (instead of mices)

....

Thanks to the substantial improvement I feel like endorsing this MS publication.

Author Response

Dear Reviewer 2 

Thank you for accepting our first revision. We have corrected the spelling errors you found. Please excuse this.

Additionally, we have revised the English and the language style of our manuscript. We have included our corrections in several passages of the manuscript.

Reviewer 3 Report

Iannis et al. are presenting a very interesting paper on the pivotal role of TNFa during muscle regeneration while providing evidences on the clinical impact of tnfa blockage as a valuabe approach to positively modulate muscle regeneration.

Minor point

The introduction digests fast the role of TNFa. The cell source(s), the cell target(s) and the biological functions must be discussed to offer to readers a suited background to better interpret the data

Major points

Authors are analyzing data coming from muscle injury on soleus muscles. Soleus is composed by oxidative fibers, biasing authors conclusions. Repeating the same experiments in muscles with a mixed composition of glycolytic and oxidative fibers will help in generalizing the results to a more neutral muscle context. Improving this part is considered crucial for the manuscript. In case of the impossibility to add new results, such limitation should clearly discussed in the conclusion senction.

The authors do not consider possible complications and do not perform any basic experiments to investigate the impact of TNFa blockage at the level of the stem cell niche (see recent review reports). Fibro/adipogenic progenitors control most aspects of the regenerative program by producing new matrix components that guide SCs to form new muscle fibers. FAP retention in the site of injury is transient and controlled by TNFa that stimulates apoptosis of excess of FAPs (Lemos 2015). Moreover, Marinkovic et al demonstrated that TNFa, via synergistic interplay between NFKB and notch, restrains FAP adipogenesis by favoring regeneration in the mdx muscles. In light of these reports, the authors should comment on the fact that TNFa inhibition may have a deleterious role due to FAP accumulation and fat infiltration. Improving this part is mandatory for publication in this journal.

Author Response

Dear Reviewer 3

We thank you for your suggestion. 

Minor point: 

"The introduction digests fast the role of TNFa. The cell source(s), the cell target(s) and the biological functions must be discussed to offer readers a suited background to better interpret the data. "

We have expanded the introduction of our manuscript and included the requested background information. Please see pages 1 and 2 line 39–53.

Major point 1: 

"Authors are analyzing data coming from muscle injury on soleus muscles. Soleus is composed by oxidative fibers, biasing authors conclusions. Repeating the same experiments in muscles with a mixed composition of glycolytic and oxidative fibers will help in generalizing the results to a more neutral muscle context. Improving this part is considered crucial for the manuscript. In case of the impossibility to add new results, such limitation should clearly be discussed in the conclusion section."

It is not possible for us to repeat the experiments because no other muscle tissue besides the soleus muscle was excised or stored for further analysis. Additionally, the muscle strength measurement was standardized by our group only for the soleus muscle. Therefore, we would prefer to discuss this issue. 

Here we include the corresponding section of our manuscript (page 12; line 526 to 537) that adresses this point:

"A limitation of our study is, that our results cannot be generalized for every peripheral muscle. Type-I fibers have a higher rate of protein turnover and are more vulnerable to atrophy after inactivity or denervation. Type-II fibers, on the other hand, are negatively affected by protein degradation (PMID: 23425621). The soleus muscle predominantly contains type-I (slow oxidative) fibers. Studies in healthy human subjects have shown that TNF-α is expressed mainly by type-II (fast oxidative and fast glycolytic) fibers (PMID: 16385844). Pathological conditions, including muscle wasting appears to be TNF-α mediated and myofiber specific. Sciorati et al. could show that the proportion of type-II fibers decreases, and that type-I fibers increase during aging-induced muscle wasting. The authors found out, that inhibition of TNF-α could reduce the loss of type-II fibers in aged animals (PMID: 33260150). We therefore assume, that the muscle regeneration after TNF-αblockade could be more pronounced in predominantly type-II muscles than in predominantly type-I muscles." 

Major point 2: 

"The authors do not consider possible complications and do not perform any basic experiments to investigate the impact of TNFa blockage at the level of the stem cell niche (see recent review reports). Fibro/adipogenic progenitors control most aspects of the regenerative program by producing new matrix components that guide SCs to form new muscle fibers. FAP retention in the site of injury is transient and controlled by TNFa that stimulates apoptosis of excess of FAPs (Lemos 2015). Moreover, Marinkovic et al demonstrated that TNFa, via synergistic interplay between NFKB and notch, restrains FAP adipogenesis by favoring regeneration in the mdx muscles. In light of these reports, the authors should comment on the fact that TNFa inhibition may have a deleterious role due to FAP accumulation and fat infiltration. Improving this part is mandatory for publication in this journal."

We thank you for your suggestion. We include a new section of our manuscript (page 12; line 501 to 515) that adresses this point:

Recent studies have identified the crucial role of fibro/adipogenic progenitors (FAPs) during muscle regeneration and repair (PMID: 31496956). FAPs are an essential source of matrix-producing myofibroblasts in the skeletal muscle and are in pathologic conditions responsible for muscle fibrosis and fatty degeneration. Some studies describe a negative effect of muscle regeneration after TNF-a inhibition. This argumentation is predominantly based on the action of FAPs and by extension on the fatty degeneration of the muscle. During repair, FAPs are involved in the muscle regeneration by creating a temporary niche that aids the myoblasts’ function. Almost immediately after muscle injury, infiltrating macrophages induce apoptosis in FAPs. This apoptotic process is mediated via TNF-signaling (PMID: 26053624). TNF blockade leads to increased survival of FAP and increased collagen deposition in the injured muscle tissue (PMID: 26053624). Marinkovic et al. provided evidence, that NOTCH- and TNF-α-signaling, control FAP differentiation in the skeletal muscle (PMID: 31239312). TNF-α, restrains FAP adipogenesis via a synergistic interaction between NF-κB and the NOTCH-signaling, by favoring regeneration in the mdx muscles. In the same study, the use of anti-TNF-α antibodies enhanced the antiadipogenic effect (PMID: 31239312). 

Round 2

Reviewer 3 Report

Authors have substantially improved the manuscript as indicated in last round of revision.